# VideoCapsuleNet: A Simplified Network for Action Detection

**Kevin Duarte**
kevin_duarte@knights.ucf.edu

**Yogesh S Rawat**
yogesh@crcv.ucf.edu

**Mubarak Shah**
shah@crcv.ucf.edu

Center for Research in Computer Vision
University of Central Florida
Orlando, FL 32816

## Abstract

The recent advances in Deep Convolutional Neural Networks (DCNNs) have shown extremely good results for video human action classification, however, action detection is still a challenging problem. The current action detection approaches follow a complex pipeline which involves multiple tasks such as tube proposals, optical flow, and tube classification. In this work, we present a more elegant solution for action detection based on the recently developed capsule network. We propose a 3D capsule network for videos, called VideoCapsuleNet: a unified network for action detection which can jointly perform pixel-wise action segmentation along with action classification. The proposed network is a generalization of capsule network from 2D to 3D, which takes a sequence of video frames as input. The 3D generalization drastically increases the number of capsules in the network, making capsule routing computationally expensive. We introduce *capsule-pooling* in the convolutional capsule layer to address this issue and make the voting algorithm tractable. The routing-by-agreement in the network inherently models the action representations and various action characteristics are captured by the predicted capsules. This inspired us to utilize the capsules for action localization and the class-specific capsules predicted by the network are used to determine a *pixel-wise localization* of actions. The localization is further improved by parameterized skip connections with the convolutional capsule layers and the network is trained end-to-end with a classification as well as localization loss. The proposed network achieves state-of-the-art performance on multiple action detection datasets including UCF-Sports, J-HMDB, and UCF-101 (24 classes) with an impressive ∼20% improvement on UCF-101 and ∼15% improvement on J-HMDB in terms of v-mAP scores.

## 1 Introduction

Human action detection is a challenging computer vision problem, which involves detecting human actions in a long video as well as localizing these actions both spatially and temporally. In recent years, great progress have been achieved in solving action detection problem using deep learning methods [1]. Although the existing approaches have achieved a reasonable performance, these methods can be very complex. These networks tend to use multi-stage pipelines, which extract action proposals from a sequence of frames, classify these regions, and perform bounding box regressions on the proposals [2, 3, 4]. The two-stream networks [5, 6] perform better but they require computation and processing of optical flow. To overcome this drawback, we propose a simpler and more elegant solution to action detection through the use of capsules.

Capsule networks were introduced in [7] for the task of image classification. A capsule is a group of neurons which can model different entities or parts of entities. The capsules in a network undergo a routing-by-agreement algorithm which enables the capsule network to build part-to-whole relationships between entities and allows capsules to learn viewpoint invariant representations. Through this improved representation learning, capsule networks are able to achieve state-of-the-art results in image domain with a drastic decrease in the number of parameters.

In this work, we aim at generalizing the capsule network from images to videos for the task of action detection. The proposed network, **VideoCapsuleNet**, uses 3D convolutions along with capsules to learn semantic information necessary for action detection. The predicted capsules can capture the visual and motion characteristics of the input video clip which helps in action recognition. The network also has a localization component which utilizes the action representation captured by the capsules for a pixel-wise localization of actions. The capability of the capsules to learn meaningful representations of actions allows the localization network to predict fine pixel-wise segmentations of actions. VideoCapsuleNet is a much simpler network which can identify and localize actions in a given video, without the need of a region proposal network or optical flow information. Furthermore, it decreases the number of network parameters by using a simple encoder-decoder architecture, which takes a video clip as input and action localization and classification as an output and is trained end-to-end.

In summary, the main contribution of this work is the proposal of 3D capsule network to solve the problem of action detection in videos. To the best of our knowledge, this is the first work on capsules in the video domain. We present a novel *capsule-pooling* procedure for capsule routing, which greatly reduces the computational cost of routing in convolutional capsule layers. The network achieves state-of-the-art action localization results on the UCF-Sports, J-HMDB, and UCF-101 datasets with ∼15-20% improvement on J-HMDB and UCF-101. Apart from action classification and pixel-wise localization, the predicted capsules in the network are also capable of explaining different characteristics of the action in the video.

## 2   Related Work

**Action Detection**   The most successful action classification methods involve the use of CNNs [1]. Earlier deep learning works used CNNs to detect human actions in each frame and then stitch these detections to create spatio-temporal tubes [8, 9]. Simonyan et al. [5] use a two-stream (spatial and temporal) CNN which processes a single frame along with multiple optical flow frames. Although the use of the temporal stream exploits motion in the video and improves accuracy, it requires a separate optical flow computations for each video. 3D CNNs [10] have been shown to successfully extract spatio-temporal features, which can be used for action classification. The 3D kernels allow the CNN to learn temporal/motion information directly from the video frames. More recently, [6] propose a two-stream I3D network which take advantage of ImageNet pretraining by inflating 2D ConvNets into 3D.

Approaches for action detection require networks to not only classify actions, but also localize them. Kalogeiton et al. [4] use 2D CNNs to extract frame-level features and create action proposals through the use of anchor cuboids. These cuboids are then classified and refined through regression. Similarly, the TCNN [2] use anchor boxes to create tube proposals, which are linked together and classified. The baseline presented in [3] extends the I3D network for action localization by having a region proposal network that selects spatio-temporal regions to be classified and refined. Although the existing work shows promising results, all these approaches require complex region proposal networks that extract and classify spatio-temporal regions. As the complexity of these networks increase, it becomes more difficult to optimize the large number of parameters.

**Capsules**   Sabour et al. [7] presented a capsule as a vector of neurons, whose orientation represents the properties of the entity and whose length represents the entity's existence. The routing algorithm measures agreement through a scalar product between two capsule vectors. In [11], Hinton et al. separate a capsule into a 4x4 pose matrix and an activation probability, to model the properties and existence of entities. The routing-by-agreement was replaced by a modified EM-algorithm which can better model the agreement between capsules. For the proposed VideoCapsuleNet, we use the capsules and routing algorithm similar to [11]. In both of above works, the capsule networks were applied to images no larger than $32 \times 32$. When dealing with larger images, or in this case videos of

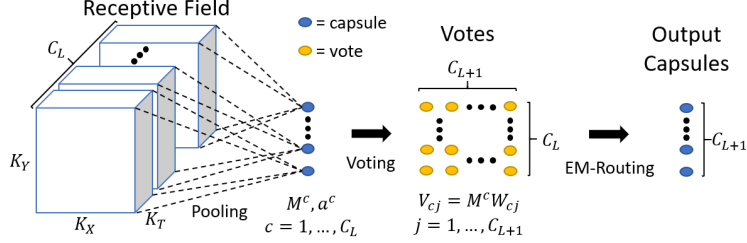

Figure 1: Capsule Pooling. New capsules are created by averaging the capsules in the receptive field for each capsule type. These new capsules then undergo the voting and routing-by-agreement procedure to obtain the capsules for the following layer.

size $8 \times 112 \times 112$, routing of capsules become computationally expensive. We address this issue by implementing a capsule-pooling procedure in convolutional capsule layers (explained in section 3.1).

## 3  Generalizing capsules to higher dimensional inputs

For human action detection in videos, it is necessary to have a large enough network to successfully model the high dimensional data. Capsule transformation and routing is computationally expensive when compared with conventional convolutions and pooling. This makes generalization of capsule network to 3D very challenging. Therefore, it is crucial to optimize the routing procedure when scaling capsule networks to high dimensional inputs like videos.

A capsule is composed of a 4x4 pose matrix, $M$, and an activation probability, $a$ [11]. The pose matrix contains the instantiation parameters, or properties, of the entity, which it models and the activation probability is a scalar between $0$ and $1$, which represents the existence of the entity. The transformation matrix, $W_{ij}$, is used by a capsule $i$ in layer $L$ to cast a vote, $V_{ij} = M_i W_{ij}$, for the pose matrix $M_j$ of a capsule $j$ in layer $L + 1$. The votes from all capsules in layer $L$ are then used in an EM routing procedure to obtain the pose matrices and activation probabilities of the capsules in layer $L + 1$. Let $N$ be the number of capsules in layer $L$, then the routing between layers $L$ and $L + 1$ requires $N_L \text{x} N_{L+1}$ votes to be computed. When the number of capsules in any layer becomes too large, the routing procedure becomes computationally intractable.

**Convolutional Capsule Routing**   Convolutional capsules reduce the number of routed capsules by only computing votes for capsules within a local receptive field. In this case, the number of votes that undergo routing is proportional to the receptive field's volume times the number of capsule types. However, this is not enough to reduce the computational cost if (i) the kernel/receptive field volume is large, as in our case when using 3-dimensional kernels, or (ii) the spatial/temporal dimensions of the convolutional capsule layer is large. In the previous 2-D capsule works for images, this is not an issue as the dimensions of the convolutional capsule layers are no larger than $14 \times 14$ and $3 \times 3$ kernels are used. When dealing with videos, these dimensions must be much larger: our first convolutional capsule layer has the dimensions $6 \times 20 \times 20$ and each capsule in the following capsule layer has a receptive field of $3 \times 5 \times 5$.

### 3.1  Capsule-Pooling

We propose a new voting procedure for convolutional capsule layers to reduce the number of computations used in capsule routing. First, we share transformation matrices between capsules of the same type; since capsules of the same type model the same entity at different positions, their votes should not vary based on their position. This decreases the number of learned parameters, which reduces the computation needed for the backward pass during training. Next, we reduce the number of votes being routed, by only applying the transformation matrix on the *mean* of the capsules in the receptive field of each capsule type.

More formally, consider convolutional capsule routing between two layers, $L$ and $L + 1$, where $C$ is the number of capsule types in a layer. For 3D convolutional capsules, the receptive field of the capsules in layer $L + 1$ has the shape $(K_T, K_X, K_Y)$. In conventional convolutional capsule routing, each capsule in the receptive field would cast $C_{L+1}$ votes, resulting in $C_L \times C_{L+1} \times K_T \times K_X \times K_Y$

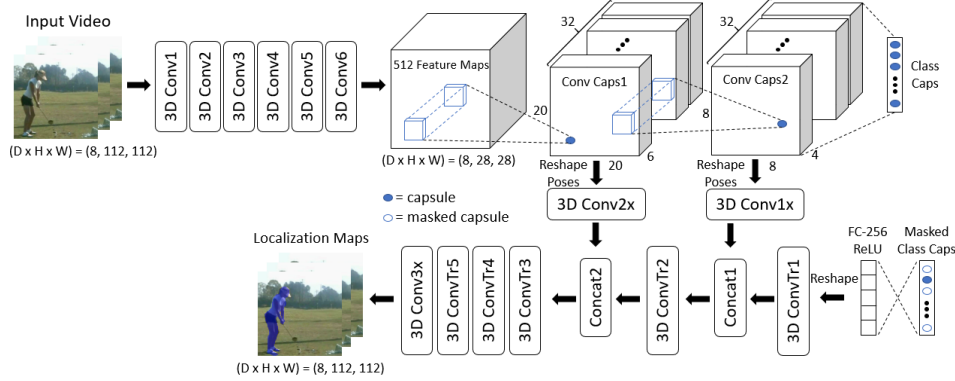

Figure 2: The VideoCapsuleNet Architecture. Features are extracted from the input frames by using 3D convolutions. These features are used to create the first capsule layer Conv Caps1. This is then followed by a convolutional capsule layer Conv Caps2 and then a fully connected capsule layer Class Caps. The decoder network uses the masked class capsules, skip connections from the convolutional capsule layers, and transposed convolutions to produce the pixel-wise action localization maps.

votes for the routing procedure at each spatio-temporal position of layer $L + 1$. Since capsules of the same type model the same entity at different positions, we can safely assume that capsules of the same type that are close to each other should have similar poses and activations. Therefore, using the same transformation matrix on each capsule within a local receptive field would result in similar votes. This means that $K_T \times K_X \times K_Y$ similar votes are calculated $C_L \times C_{L+1}$ times. Each of these similar votes adds little useful information to the routing algorithm, making them redundant and unnecessary to compute. Instead of computing these redundant votes, we implement a capsule-pooling procedure as shown in Figure 1. For each capsule type, $c$, in layer $L$, we create one capsule with a pose matrix $M^c$ and an activation $a^c$ as follows:

$$M^c = \frac{1}{K_T K_X K_Y} \sum_{k=1}^{K_T} \sum_{i=1}^{K_X} \sum_{j=1}^{K_Y} M_{kij}^c, \quad a^c = \frac{1}{K_T K_X K_Y} \sum_{k=1}^{K_T} \sum_{i=1}^{K_X} \sum_{j=1}^{K_Y} a_{kij}^c, \tag{1}$$

where $M_{kij}^c$ and $a_{kij}^c$ are the pose matrix and activation of the capsule at position $(k, i, j)$ in the receptive field. Now, each one of these capsules casts a vote for each capsule type in the layer $L + 1$, resulting in a total of $C_L \times C_{L+1}$ votes. Thus, capsule-pooling ensures we do not compute many similar votes; it ensures that the number of votes is only proportional to the number of capsule types in each layer, and indifferent to the volume of the receptive field.

## 4 Network Architecture

The VideoCapsuleNet architecture is shown in Figure 2. The input to the network is 8 $112 \times 112$ frames from a video. The network begins with 6 $3 \times 3 \times 3$ convolutional layers (each with ReLU activations) which result in 512 feature maps of dimension $8 \times 28 \times 28$. The first capsule layer is composed of 32 capsule types. The capsule 4x4 pose matrices and activations are obtained by applying a $3 \times 9 \times 9$ convolution operation, with ReLU and sigmoid activations respectively, to these 512 feature maps. This is followed by a second convolutional capsule layer with 32 capsule types, a $3 \times 5 \times 5$ receptive field, and a stride of $1 \times 2 \times 2$.

This second, and final, convolutional capsule layer is then fully connected to $C$ capsules, where $C$ is the number of action classes. For this final classification layer (class capsules), the capsule with the largest activation corresponds to the network's action prediction. When computing the votes for this final convolutional capsule layer, all capsules of the same type share transformation matrices. In order to preserve the information about the convolutional capsules' locations, we perform Coordinate Addition [11]: at each position, we add the capsules' coordinates (time, row, column) to the final three entries of the vote matrix.

**Localization Network**   To obtain frame-level action localizations, we want to leverage the action-based representation found in the class capsule layer's pose matrices. To this end we use the masking

procedure as follows. During training we mask all pose matrices except for the one corresponding to the ground truth class, by setting their values to zero. At test time, all class capsules except the one with the largest activation, the predicted action, are masked. The class capsule poses are then fed into a fully connected layer which produces a $4 \times 8 \times 8$ feature map. This feature map corresponds to a rough localization of the action in the video. The localization is then upscaled through a series of transposed convolutions that result in 8 $112 \times 112$ localization maps. To ensure fine positional information is incorporated in this final localization, skip connections are used from the convolutional capsule layers; the pose matrices of these capsule layers are flattened and are used in a conventional convolution layer. Their outputs are then concatenated with the transposed convolution outputs.

## 4.1 Objective Function

VideoCapsuleNet is trained end-to-end using an objective function which is the sum of two losses: a classification loss and a localization loss. We use spread loss for classification which is computed as,

$$L_c = \sum_{i \neq t} \max(0, m - (a_t - a_i))^2, \tag{2}$$

where, $a_i$ is the activation of the final class capsule corresponding to capsule $i$, and $a_t$ is the target class' activation. The margin $m$ is linearly increased from $0.2$ to $0.9$ during training.

The network predicts a set of segmentation maps for action localization and sigmoid cross entropy is used to compute the loss. The shape of the network prediction is $(T, X, Y)$, where $T$ corresponds to the temporal length, $X$ corresponds to the height, and $Y$ corresponds to the width of the prediction volume. The posterior probability of a pixel at position $(k, i, j)$ of the predicted volume for an input video $\hat{v}$ can be expressed as,

$$p_{kij} = \frac{e^{F_{kij}(\hat{v})}}{1 + e^{F_{kij}(\hat{v})}}, \tag{3}$$

where, $F_{kij}$ is the activation value for pixel at position $(k, i, j)$ of the predicted volume for an input video $\hat{v}$. The ground truth bounding box for a video is used to assign a actionness score (0 or 1) to each pixel position in the video. Let the ground truth actionness score of a pixel at position $k, i, j$ in the input video $\hat{v}$ is defined as $\hat{p}_{kij}$, then the cost function to be minimized for action localization is,

$$L_s = -\frac{1}{TXY} \sum_{k=1}^{T} \sum_{i=1}^{X} \sum_{j=1}^{Y} [\hat{p}_{kij} log(p_{kij}) + (1 - \hat{p}_{kij}) log(1 - p_{kij})]. \tag{4}$$

Thus, VideoCapsuleNet is trained using the objective function, $L = L_c + \lambda L_s$, where, $\lambda$ is used to down-weight the localization loss so that it does not dominate the classification loss. In all experiments, we use $\lambda = 0.0002$.

## 5 Experiments

**Implementation Details** We implement VideoCapsuleNet using Tensorflow [12]. For all experiments, the first 6 conv layers use C3D [10] weights, pretrained on the Sports-1M [13]. The network was trained using the Adam optimizer [14], with a learning rate of $0.0001$. Due to the size of the VideoCapsuleNet, a batch size of 8 was used during training. We measure the performance of our network on three datasets UCF-Sports [15], J-HMDB [16], UCF-101 [17]. The only video preprocessing used is the downsampling of each video such that their shortest side is 120 px. We randomly crop 112x112 patches from 8 frame video during training and take a center crop at test time. For UCF-Sports and UCF-101, we consider all pixels within the bounding box to be the ground-truth foreground while pixels outside of the bounding box are considered background. This results in more box-like segmentations, but in many cases VideoCapsuleNet produces tighter segmentations around the actor than the ground-truth bounding boxes (Figure 3).

**Metrics** We compute frame-mAP and video-mAP for the evaluation [8]. For frame-mAP we set the IoU threshold at $\alpha = 0.5$, and compute the average precision over all the frames for each class. This is then averaged to obtain the f-mAP. For video-mAP the average precision is computed for the 3D IoUs at different thresholds over all the videos for each class, and then averaged to obtain the v-mAP.

Table 1: Action localization accuracy of VideoCapsuleNet. The results reported in the row VideoCapsuleNet* use the ground-truth labels when generating the localization maps, so they should not be directly compared with the other state-of-the-art results.

| Method | UCF-Sports | | J-HMDB | | UCF-101 | | | | |
|---|---|---|---|---|---|---|---|---|---|
| | f-mAP | v-mAP | f-mAP | v-mAP | f-mAP | | | v-mAP | |
| | 0.5 | 0.2 | 0.5 | 0.2 | 0.5 | 0.1 | 0.2 | 0.3 | 0.5 |
| Saha et al. [18] | - | - | - | 72.6 | - | 76.6 | 66.8 | 55.5 | 35.9 |
| Peng et al. [8] | 84.5 | 94.8 | 58.5 | 74.3 | 65.7 | 77.3 | 72.9 | 65.7 | 35.9 |
| Singh et al. [19] | - | - | - | 73.8 | - | - | 73.5 | - | 46.3 |
| Kalogeiton et al. [4] | 87.7 | 92.7 | 65.7 | 74.2 | 69.5 | - | 77.2 | - | 51.4 |
| Hou et al. [2] | 86.7 | 95.2 | 61.3 | 78.4 | 67.3 | 77.9 | 73.1 | 69.4 | - |
| Gu et al. [3] | - | - | 73.3 | - | 76.3 | - | - | - | 59.9 |
| He et al. [20] | - | 96.0 | - | 79.7 | - | - | 71.7 | - | - |
| VideoCapsuleNet | 83.9 | **97.1** | 64.6 | **95.1** | **78.6** | **98.6** | **97.1** | **93.7** | **80.3** |
| VideoCapsuleNet* | 82.8 | 97.1 | 66.8 | 95.4 | 80.1 | 98.9 | 97.4 | 94.2 | 82.0 |

## 5.1 Results

**UCF-Sports and J-HMDB**    The UCF-Sports dataset consists of 150 videos from 10 action classes. All videos contain spatio-temporal annotations in the form of frame-level bounding boxes and we follow the standard training/testing split used by [21]. The J-HMDB dataset contains 21 action classes with a total of 928 videos. These videos have pixel-level localization annotations. Due to the size of these datasets, we pretrain the network using the UCF-101 videos, and fine-tune on their respective training sets. On UCF-Sports, we observe a slight improvement ($\sim$1%) in terms of v-mAP (Table 1). On J-HMDB, VideoCapsuleNet achieves a 15% improvement in v-mAP with a threshold of $\alpha = 0.2$ (Table 1). In both of these datasets, we find that we do not outperform the state-of-the-art when the f-mAP or v-mAP IoU thresholds are large. We attribute this to the small number of training videos per class (about 10 for UCF-Sports and about 30 for J-HMDB). The f-mAP and v-mAP accuracy for different thresholds can be found in the supplementary file.

**UCF-101**    Our UCF-101 experiments are run on the 24 class subset consisting of 3207 videos with bounding box annotations provided by [19]. On UCF-101 VideoCapsuleNet outperforms existing methods in action localization, with a v-mAP accuracy 20% higher than the most state-of-the-art methods (Table 1). This shows that VideoCapsuleNet performs exceptionally well when the dataset is sufficiently large.

**v-mAP and f-mAP Improvements**    In UCF-101 and J-HMDB, VideoCapsuleNet is able to greatly outperform other methods in terms of v-mAP, but does not have this large corresponding increase in f-mAP score. Current SOTA methods usually localize actions at a frame level: a region proposal network generates bounding box proposals, which are then linked together over time and regressed to improve results. These frame level predictions might produce good f-mAP results, but these proposals would not necessarily be temporally consistent. VideoCapsuleNet, on the other hand, generates segmentations for all the frames in a clip simultaneously, resulting in a more temporally consistent segmentation and an improved v-mAP score.

## 5.2 What class capsules learn?

Since all but one class capsule is masked out when the class capsules are passed to the localization network, each class capsule should contain localization information specific to their corresponding action (i.e. class capsule for *diving* should have information which would be useful when localizing the diving action). We found that this was indeed the case; at test time we masked all class capsules except the one corresponding to the ground-truth action, and localized the actions. These localization results can be found in Table 1 under VideoCapsuleNet*. When given the correct action to localize, VideoCapsuleNet is able to improve its localizations. Figure 4 shows several examples of localizations, when different class capsules are masked.

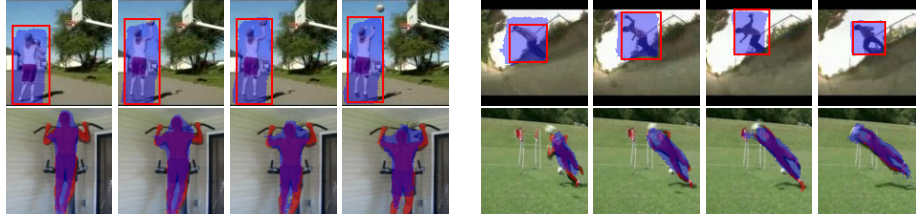

Figure 3: Sample action localizations for UCF-101 (first row) and J-HMDB (second row). The UCF-101 videos have bounding box annotations (shown in red) and the predicted localizations are in blue. J-HMDB has pixel-wise annotations (shown in red) and the predicted localizations are in blue.

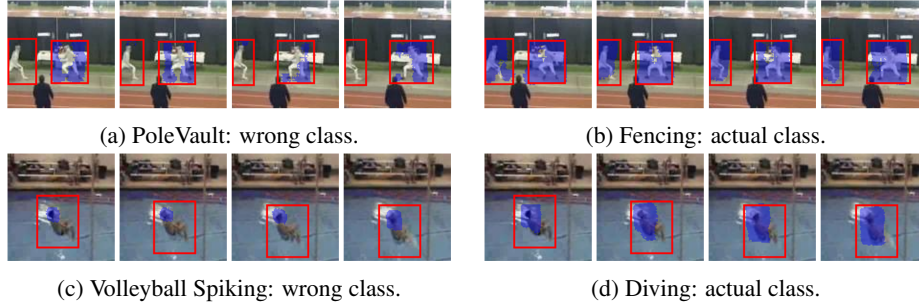

    (a) PoleVault: wrong class.                       (b) Fencing: actual class.

    (c) Volleyball Spiking: wrong class.                (d) Diving: actual class.

Figure 4: Sample localizations for UCF-101 videos (ground truth is red bounding box). The localizations (a) and (c) mask out all class capsules except the one corresponding to an incorrect action; the localizations (b) and (d) mask all capsules except the one corresponding to the correct (ground-truth) action. These localizations show that the class capsules contain action specific information and this information propagates to the localizations.

## 5.3 Ablation Experiments

**Video Reconstructions** Reconstruction can act as a regularizer in network training [7]. To this end, we perform two experiments where the network reconstructs the original video; we add a convolutional layer to 3D ConvTr5, that has 3 channel outputs to reconstruct the input video. In the first experiment, the network is trained using the sum of the classification, localization, and reconstruction losses. In the second experiment the network is trained with only the classification and reconstruction losses. These experiments show us that the addition of a reconstruction network, when no localization information are available, do help the capsules learn better representations: there is a 10% increase in performance (Table 2). However, localization information allows the capsules to learn better representations, allowing for improved classification performance. Using both the reconstruction and localization losses decrease the classification performance. We believe this additional loss forces the capsules to learn non-semantic information (RGB values), which hurts their ability to learn from the highly semantic bounding-box annotations.

**Additional Skip Connections** Due to the first 6 convolutional layers (two of which have strides of 2 in the spatial dimensions) the network may lose some spatial information; we test the effectiveness of adding skip connections from these layers. For this experiment, we add skip connections at layers 3D Conv1, 3D Conv2, and 3D Conv4 to preserve the spatial information that is lost through striding. These additional skip connections result in similar classification and localization results as the base VideoCapsuleNet (Table 2), but they increase the number of network parameters as well as the training time. For this reason, VideoCapsuleNet only has skip connections at the convolutional capsule layers.

**Coordinate Addition** Coordinate Addition allows the class capsules to encode positional information about the actions which they represent, by adding the capsules' coordinates (time, row, column) to the vote matrices of the final convolutional capsule layer. In our synthetic dataset experiments, we show that this is the case: these three capsule dimensions change predictably as the direction and

Table 2: All ablation experiments are run on UCF-101. The f-mAp and v-mAP use IoU thresholds of $\alpha = 0.5$. ($L_c$:classification loss, $L_s$:localization loss, $L_r$: reconstruction loss, SC:skip connections, NCA:no coordinate addition, 4Conv:4 convolution layers, 8Conv: 8 convolution layers, and Full: the full network.) Unless specified, the network uses only the classification and localization losses.

| | $L_c$ | $L_s$ | $L_c + L_r$ | $L_c + L_s + L_r$ | SC | NCA | 4Conv | 8Conv | Full |
|---|---|---|---|---|---|---|---|---|---|
| Accuracy | 62.0 | - | 72.2 | 73.6 | 78.7 | 71.7 | 74.6 | 71.4 | 79.0 |
| f-mAP | - | 51.1 | - | 77.8 | 77.4 | 72.9 | 72.1 | 70.4 | 78.6 |
| v-mAP | - | 48.1 | - | 79.9 | 80.7 | 74.9 | 73.5 | 71.3 | 80.3 |

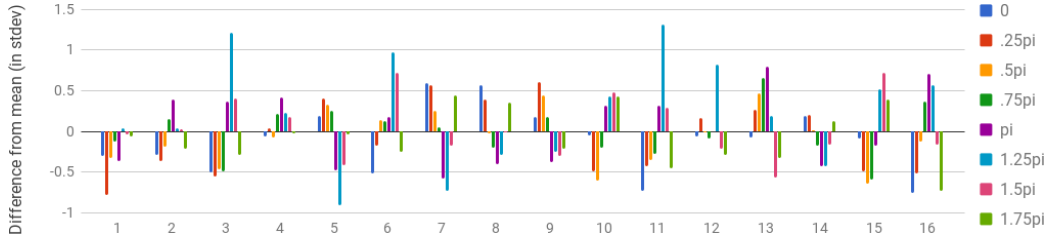

Figure 5: The 16 capsule dimensions of the Linear Motion pose matrix when the direction of motion is varied in synthetic videos. The direction 0: rightward movement, 0.25pi:diagonal movement (down and to the right), 0.5pi:downward movement. The rest of the directions follow this pattern (step of .25pi in angle). Most dimensions have a sinusoidal pattern as the direction of motion varies, which show that the pose matrix values change smoothly as video inputs change.

speed of the motion change. This improved encoding improves the networks classification accuracy by about 7% and the localization accuracy by about 5% on the UCF-101 dataset as seen in Table 2.

## 5.4 Synthetic Dataset Experiments

We run several experiments on a synthetic video dataset to better understand the instantiation parameters encoded in the class capsules' pose matrices. We use synthetic data (more details in supplementary file), since they allow us to control specific properties of the videos, which would be difficult to do with real-world videos. There are 4 action classes which corresponds to different types of motion: linear, circular, a turn, and random. VideoCapsuleNet is trained on these randomly generated videos, and then we measure the dimensions of the class capsules' pose matrices when varying different properties of the generated videos.

We found that VideoCapsuleNet's class capsules are able to parameterize the different visual and motion properties in video. Since the network uses Coordinate Addition, the final three dimensions of the pose matrices contain information about the actor's position. As we linearly increase the object's speed in the video, the dimension corresponding to the time coordinate increases in a linear fashion. Similarly, the dimensions corresponding to the row and column coordinates changes as the direction of the motion changed: vertical motion changed the dimensions corresponding to the row; horizontal motion changes the dimension corresponding to the column. This change is illustrated in the last two dimensions of Figure 5. Interestingly, these are not the only dimensions which smoothly change as the direction or speed change. Almost all capsule dimensions, for the linear motion class capsule, change smoothly as different properties (size, direction, speed, etc.) change in the video.

Since the dimensions do not change in an arbitrary fashion as the inputs change, VideoCapsuleNet's class capsules successfully encode the visual and motion characteristics of the actor. This helps explain why VideoCapsuleNet is able to achieve such good localization results; the capsules learn to represent the different spatio-temporal properties necessary for accurate action localizations.

## 5.5 Computational Cost and Training Speed

Although capsule networks tend to be computationally expensive (due to the routing-by-agreement), capsule-pooling allows VideoCapsuleNet to run on a single Titan X GPU using a batch size of 8. Also, VideoCapsuleNet trains quickly when compared to other approaches: on UCF-101 it converges in fewer than 120 epochs, or 34.5K iterations. This is substantially fewer iterations than the 70K iterations for [8], 100K iterations for the TCNN [2], 600K-1M iterations for [3].

# 6 Conclusion and Future Work

In this work we propose VideoCapsuleNet, a generalization of capsule network from 2D images to 3D videos, for action detection. To the best of our knowledge, this is the first work where capsules are employed for videos. The proposed network takes video frames as input and predicts an action class as well as a pixel-wise localization for the input video clip. We introduce capsule-pooling to optimize the voting algorithm in the convolutional capsule layers which makes the routing feasible. The proposed network has a localization component which generates pixel-wise localization considering the predicted class-specific capsules. VideoCapsuleNet can be trained end-to-end and we obtain state-of-the-art performance on multiple action detection datasets. Research on capsules is still at an initial stage and we have already seen good performance on different tasks. The basic idea behind capsule is very intuitive and there are many fundamental reasons which make capsules a better approach than conventional ConvNets, however, it will require a lot more effort to fully validate these facts. The results we have achieved in this paper on videos are promising and indicate the potential of capsules for videos, which makes it worth exploring.

### Acknowledgments

This research is based upon work supported by the Office of the Director of National Intelligence (ODNI), Intelligence Advanced Research Projects Activity (IARPA), via IARPA R&D Contract No. D17PC00345. The views and conclusions contained herein are those of the authors and should not be interpreted as necessarily representing the official policies or endorsements, either expressed or implied, of the ODNI, IARPA, or the U.S. Government. The U.S. Government is authorized to reproduce and distribute reprints for Governmental purposes notwithstanding any copyright annotation thereon.

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
