[Reviews · NeurIPS 2018]

Reviewer 1



Authors present a new method for action detection in video based on capsule networks. The pose parameters provided in capsule layers allows the network to efficiently segment the image at pixel level. They also propose averaging the convolutional capsule votes to increase the scalability of capsule networks. The effect on the generalization due to parameter reduction however is not explored in the ablation study. I am curious about the effect of capsule pooling on the results. Pros: Significant results, clear and well written paper. Cons: not enough elaboration on the discrepancy between fmap and vmap. The fmap is not better than sota, where the vmap indicates a clear outperform. I would also suggest adding the prediction of other methods in the qualitative results to indicate the behavioural difference that would lead to the 20% improvement. Moreover, the coordinate adjustment technique should also be applied in the capsule pooling section. Which means that before averaging, the pose matrices should be adjusted based on their relative position to the center of the patch.

Reviewer 2



The paper presents a method for video action detection. The proposed method extends the recently proposed 2D capsule networks to 3D. The proposed network consists of a 3D encoder (convolutional layers) followed by two capsule networks; the last of which is fully connected to C capsules (C the number of action classes). For localization, the wrong outputs of the capsules (the ones not belonging to the ground truth class) are masked out and the one remaining (that corresponds to the ground truth class) is first fed to a fully-connected layer and then to a series of transposed convolutions to produce the final pixel-level localization. The method is evaluated on three popular action localization datasets achieving state-of-the-art results. The paper is well-written and easy to follow. It tackles the difficult problem of action detection, being able to localize actions not only with bounding boxes but also with pixel-wise localizations. The results of Table 1 show that using capsules really outperforms the state of the art for all datasets. The method is mostly a simple extension of 2D capsules to 3D. Besides this extension, the main difference from the original work is capsule pooling (Section 3.1). This seems a bit contradictory to the initial capsule works and to the whole idea behind capsules [7,11], where the authors state that pooling throws away information about the precise position of the entity within a region. If there was no memory limitation, do the authors think that the network without the pooling operation could perform better? For capsule pooling, the authors use the mean of the capsules in the receptive field of each capsule type. Did the authors try to experiment with other types of pooling, such as max-pooling? It would also be interesting to see where the remainder of the error comes from, eg. if it is due to incorrect localization and/or classification of the actions. In lines 33-37 of the supplementary material, the authors attribute the low performance of some classes to different backgrounds or to cluttered scenes, which is not the case of other classes, such as cliff diving. It would be beneficial if the authors could comment on that. Moreover, It would be interesting to see what happens when there are multiple people in the scene, eg. ice dancing, with some examples or analysis maybe (given that Table 5 of the supplementary material shows that the proposed method achieves 100% video-mAP for this class). The authors do not use any motion information. It would be interesting to see how the method would perform with optical flow as input, especially for the failure cases of RGB inputs, eg. cluttered background. Minor: - line 78: work instead of works. - line 263: contain instead of contains. - Section 5.3 should also mention that the results are reported in Table 2. - A small part of the right side of Figure 2 is missing. [About author feedback] The authors did not really explain the difference between f-map and v-map. Works like [2] and [4] also use multiple frames as input and the difference between their f-map and v-map is expected (temporally consistent detections vs per-frame detections). I am not convinced by the explanation provided by the authors. Regarding the “remainder of the error”: localization can mean temporal or spatial; hence, an explanation for this could be useful. Overall, I think that it’s a good submission and I stand by my initial score.

Reviewer 3



This paper generalizes the capsule network from images to videos for the task of action detection. Strengths: The motivation of the paper is clear, and the experimental results is promising. The estimated localization maps look accurate and sharp. Weakness: 1. This paper is a simple extension of capsule network, from 2D to 3D. It is interesting and straightforward, but novelty is limited. 2. Performance in table 1 is confusing. The f-mAP is quite inconsistent with v-mAP . I don't understand how your v-mAP is so high but your f-mAP is so low. It is counter-intuitive.